# The Future of Direct Cardiac Reprogramming: Any *GMT* Cocktail Variety?

**DOI:** 10.3390/ijms21217950

**Published:** 2020-10-26

**Authors:** Leyre López-Muneta, Josu Miranda-Arrubla, Xonia Carvajal-Vergara

**Affiliations:** Regenerative Medicine Program, Center for Applied Medical Research (CIMA), Instituto de Investigación Sanitaria de Navarra, University of Navarra, 31009 Pamplona, Spain; llopez.13@alumni.unav.es (L.L.-M.); jmiranda.2@alumni.unav.es (J.M.-A.)

**Keywords:** direct reprogramming 1, iCMs (induced cardiomyocytes) 2, iCPCs (induced cardiac progenitor cells) 3, cardiovascular diseases 4, cardiovascular repair 5, cardiovascular regeneration 6

## Abstract

Direct cardiac reprogramming has emerged as a novel therapeutic approach to treat and regenerate injured hearts through the direct conversion of fibroblasts into cardiac cells. Most studies have focused on the reprogramming of fibroblasts into induced cardiomyocytes (iCMs). The first study in which this technology was described, showed that at least a combination of three transcription factors, GATA4, MEF2C and TBX5 (GMT cocktail), was required for the reprogramming into iCMs in vitro using mouse cells. However, this was later demonstrated to be insufficient for the reprogramming of human cells and additional factors were required. Thereafter, most studies have focused on implementing reprogramming efficiency and obtaining fully reprogrammed and functional iCMs, by the incorporation of other transcription factors, microRNAs or small molecules to the original GMT cocktail. In this respect, great advances have been made in recent years. However, there is still no consensus on which of these GMT-based varieties is best, and robust and highly reproducible protocols are still urgently required, especially in the case of human cells. On the other hand, apart from CMs, other cells such as endothelial and smooth muscle cells to form new blood vessels will be fundamental for the correct reconstruction of damaged cardiac tissue. With this aim, several studies have centered on the direct reprogramming of fibroblasts into induced cardiac progenitor cells (iCPCs) able to give rise to all myocardial cell lineages. Especially interesting are reports in which multipotent and highly expandable mouse iCPCs have been obtained, suggesting that clinically relevant amounts of these cells could be created. However, as of yet, this has not been achieved with human iCPCs, and exactly what stage of maturity is appropriate for a cell therapy product remains an open question. Nonetheless, the major concern in regenerative medicine is the poor retention, survival, and engraftment of transplanted cells in the cardiac tissue. To circumvent this issue, several cell pre-conditioning approaches are currently being explored. As an alternative to cell injection, in vivo reprogramming may face fewer barriers for its translation to the clinic. This approach has achieved better results in terms of efficiency and iCMs maturity in mouse models, indicating that the heart environment can favor this process. In this context, in recent years some studies have focused on the development of safer delivery systems such as Sendai virus, Adenovirus, chemical cocktails or nanoparticles. This article provides an in-depth review of the in vitro and in vivo cardiac reprograming technology used in mouse and human cells to obtain iCMs and iCPCs, and discusses what challenges still lie ahead and what hurdles are to be overcome before results from this field can be transferred to the clinical settings.

## 1. Background

### 1.1. Cardiogenesis

The mammalian heart is the first organ formed in the developing embryo and is composed of four chambers (right atrium, right ventricle, left atrium, and left ventricle) and three layers: the endocardium (the innermost endothelial layer), the myocardium (the middle muscular layer), and the epicardium (the outermost mesothelial layer). The function of the heart as a pump for blood is essential for both the proper circulation of nutrients and oxygen and the removal of metabolic waste, and this mechanical beating is coupled to electrical signals from the cardiac conduction system. A healthy human heart pumps approximately 5 L of blood every minute, 7500 L per day. Therefore, the formation of the complex structure of the heart, is orchestrated by the integration and contribution of different progenitor populations from the cardiogenic mesoderm, pro-epicardium, and neural crest in response to different spatio-temporal signals and a tight control of specific gene regulatory networks [1,2,3]. The complexity of heart morphogenesis is shown by the wide spectrum of congenital heart defects that become life-threatening cardiac anomalies at birth or later during adult life.

Early during the first stages of cardiac development, cells from the lateral anterior splanchnic mesoderm migrate to the cranio-lateral region of the embryo to form the anterior lateral plate, a cardiogenic area where cardiac progenitor cells (CPCs) are formed. Different signals regulate the commitment of these cardiogenic mesodermal precursors to two distinct CPCs called first heart field (FHF) and second heart field (SHF). The CPCs in the FHF migrate medially from the cardiogenic area and form the cardiac crescent from around murine embryonic day (E) 7.5, human E14, and start to be specified and differentiate into cardiomyocytes (CMs) prior to CPCs in the SHF. These cells differ in their molecular signature, location in the cardiac crescent and contribution to the developing heart structures. FHF gives rise to the left ventricle, part of the interventricular septum, part of the atria and a minor part of the right ventricle. SHF gives rise to a majority of cells of the right ventricle, the inflow and outflow tracts, and part of the atria [3]. Interestingly, in the part of the SHF called the anterior heart field (AHF), CPCs marked by the activation of *Mef2c*-AHF enhancer [4], reside within the pharyngeal mesoderm during early cardiogenesis. At this stage and in this particular niche, these AHF CPCs are mulitpotent and can proliferate through the expression of N-cadherin and interaction with canonical Wnt signals [5] before they migrate and enter the developing primitive heart tube and differentiate into the variety of cell lineages (including CMs, vascular endothelial cells, smooth muscle cells, and fibroblasts) that form the outflow track and right ventricle [6,7,8]. One of the few gene markers of FHF is *Hcn4*, which encodes hyperpolarization-activated cyclic nucleotide gated potassium-channel 4 [9]. In contrast, several specific markers of the SHF such as FGF10, FGF8, ISL1, TBX1, MEF2C-AHF, and SIX2 have been found [6].

In terms of cellular composition, the major cell types that form the heart are CMs, cardiac fibroblasts, vascular smooth muscle cells (localized fundamentally within the myocardium) and vascular endothelial cells (found within the myocardium and endocardium) [10,11]. CMs, are specialized cells with a complex filament structure responsible for control of the rhythmic beating of the heart. CMs are mainly derived from FHF and SHF [1]. CMs present heterogeneity depending on the location, morphology, and function, including atrial, ventricular, sinoatrial nodal, atrioventricular nodal, His bundle, and Purkinje fibers [12,13]. The most common pan-markers used to detect CMs are myosin heavy chain (MHC) and cardiac troponin (cTn). MHC forms part of type II myosin, one of the major components of sarcomeres, whereas cTn proteins control the calcium mediated interaction between actin and myosin. The expression of the different isoforms of these proteins (αMHC or βMHC; cTnT or cTnI) depends on the developmental stage and heart compartment (atrial or ventricular) and can be affected by physiological and pathologic conditions [14,15], although more specific markers can define each CM subtype [12].

From a genetic point of view, there are specific genes expressed in early stages of cardiogenesis that play a pivotal role in heart development [2,3,16,17]. Specifically:MEF2. MEF2 is a MAD-box containing transcription factor with a key role in heart morphogenesis and in the regulation of the CPC and CM gene program [4,18]. MEF2 is encoded by four genes, *Mef2a*, *-b*, *-c*, and *-d*. *Mef2b* and *Mef2c* are the first MEF2 isoforms expressed in the cardiac mesoderm at mouse E7.5, *Mef2a* and *Mef2d* are expressed in the linear heart tube between E8.0 and E8.5, and after E8.5, all four *Mef2* genes are expressed throughout the developing heart [18]. *Mef2c* is required for activation of a subset of cardiac contractile protein genes, as well as for the development of cardiac structures derived from SHF [4]. In mice homozygous for a null mutation of *Mef2c*, the heart tube did not undergo looping morphogenesis, the future right ventricle did not form, and a subset of cardiac muscle genes was not expressed [19].HAND2. *Hand1 and Hand2* encode basic helix-loop-helix transcription factors and are expressed in mesodermal and neural crest-derived structures of the developing heart. *Hand2* is expressed in the outflow track, the epicardium, valve progenitors, and predominantly in the myocardial compartment of the right ventricle, while the related transcription factor *Hand1* is predominantly expressed in the left ventricle [20,21]. Deletion of *Hand2* results in severe hypoplasia of the right ventricle segment [22]. In fact, the absence of the right ventricular region of *Mef2c* mutant correlated with downregulation of the HAND2 [19]. HAND2 interacts with non-coding regions of many genes involved in cardiogenesis [21].GATA4. The *Gata4* gene is expressed in CMs and their mesodermal precursors, as well as in the endocardium and the epicardium. GATA4 regulates expression of myocardium-related genes and is necessary for the proliferation of CMs, formation of the endocardial cushions, development of the right ventricle and septation of the outflow tract [23]. GATA4 binds and promotes deposition of H3K27ac, and subsequently, establish active chromatin regions, at multiple cardiac enhancers to stimulate transcription [24].BAF60c. *Smarcd3* gene, encodes BAF60c, a cardiac-enriched subunit of the SWI/SNF-like BAF chromatin complex. BAF60c is expressed specifically in the heart and somites in the early mouse embryo. *Smarcd3* silencing in mouse embryos causes defects in heart morphogenesis that reflect impaired expansion of the AHF, and results in abnormal cardiac and skeletal muscle differentiation [25]. Baf60c regulates a gene expression program that regulates the main functional properties of CMs, including genes encoding contractile proteins, modulators of sarcomere function, and cardiac metabolic genes. Interestingly, many of the genes deregulated in Baf60c null embryos are targets of the *MYOCD*, another important transcription factor in heart development [26], which can functionally interact with BAF60c [27].TBX5. *Tbx5* gene is a T-box transcription factor, expressed early in development throughout the entire cardiac crescent. Lineage tracing of *Tbx5* showed that this gene is expressed in the myocardium of the left ventricle, but not the right ventricle or outflow track, besides a population of the posterior SHF (contributing to the myocardium of the atria and the venous pole) [28]. TBX5 can have both positive and negative transcriptional activity depending on the transcription factors with which it interacts [29]. Interestingly, in 2009 Takeuchi et al. demonstrated the transdifferentiation of mouse mesoderm into beating CMs by the ectopic expression of GATA4, BAF60c, and TBX5. The authors described that BAF60c enabled binding of GATA4 to cardiac genes to initiate the cardiac expression program, whereas TBX5 repressed noncardiac mesodermal genes and promoted differentiation into CMs [30].NKX2.5. *Nkx2.5* gene is a homeobox transcription factor essential for early heart formation. *Nkx2.5* knockout mice die at E9.5-10.5 with severely underdeveloped heart [31]. NKX2.5 is expressed in the cardiac crescent stage and regulates CM differentiation [32]. Interestingly, NKX2.5 is expressed at lower levels in SHF progenitors than in FHF progenitors and CMs, and its expression level, combined with other factors, may trigger different outcomes. In SHF progenitors, NKX2.5 can promote proliferation and activate the expression of *Fgf10* and *Mef2c*-AHF enhancer, together with FOXH1, whereas in the FHF, NKX2.5 reduces *Fgf10* and *Isl1* expression and induces differentiation [33].MESP1. *Mesp1* is a basic helix-loop-helix transcription factor expressed in early mesoderm during gastrulation by the T-box transcription factor EOMES in response to low doses of NODAL/SMAD2/3 signaling [34]. MESP1 expressing cells migrate out from the primitive streak and are incorporated into the heart field to generate a single heart tube [35]. MESP1 acts as a master regulator of multipotent CPCs specification. It activates many genes that form the core cardiac transcriptional machinery and represses the expression of genes that control other early mesoderm and endoderm cell fates [36].

As described further below, the aforementioned genes have been considered and used in direct cardiac reprogramming strategies towards CM- and CPC-like states.

### 1.2. Regenerative Medicine to Treat Cardiac Diseases: Where Do We Stand?

During mammalian embryogenesis, CMs can proliferate to support heart growth, but CMs become terminally differentiated shortly after birth and mostly lose the ability to proliferate. Despite the long-standing concept that human heart CMs exit the cell cycle after birth and are unable to renew, there is evidence that CMs can slowly self-renew although the data on CM turnover are controversial. The integration of ^14^C from nuclear fallout into DNA allowed the age of CMs to be established, with the finding that human CMs turnover gradually decreases throughout life to less than 1% per year in adulthood [37]. On the other hand, some reports have supported the idea that CPCs may exist in the adult mammalian heart. ISL1+ CPCs have been observed in the post-natal human myocardium, but there is no evidence that they still exist in adult life [38]. Clusters of ISL1+ cells have been found in the adult murine heart, but the authors concluded that ISL1 could be considered a novel marker of the adult sinoatrial node, but these cells did not function as embryonic CPCs [39]. c-KIT expressing cells have also been found in the adult human heart [40,41], but it was demonstrated that these cells do not have a cardiac but hematopoietic origin [42], and the veracity of some published results regarding these cells has recently been questioned [43]. Thus, although many adult CPCs have been described in the literature [44], none of these have been demonstrated to be remnant and have the great cardiomyogenic potential of embryonic cardiogenic mesodermal precursors.

In any case, the real evidence is that the mammalian heart lacks the ability to efficiently and sufficiently replace the large loss of CMs following heart injury, in contrast to the observed heart regeneration in organisms such as zebrafish and newts [45,46].

An estimated 17.9 million people died from cardiovascular diseases in 2016, representing 31% of all global deaths. By 2030, 23.6 million people are estimated to die annually from cardiovascular diseases. Heart attacks and strokes, which are mainly caused by a blockage of blood flow to the heart or brain, represent 85% of these deaths (https://www.who.int/news-room/fact-sheets/detail/cardiovascular-diseases-(cvds)).

Acute myocardial infarction (AMI) occurs when coronary blood flow is decreased, resulting in acute reduction of the blood supply (ischemia) to a portion of the myocardium. Following an AMI, the damage becomes irreversible, tissue necrosis is triggered, and up to 1 billion cardiac cells are lost. Over time, AMI is followed by a remodeling process, and the damaged tissue is fundamentally substituted by a fibrotic scar and is compensated through the hypertrophy of the remaining and surrounding CMs [47]. Ventricular remodeling entails the thinning of the wall and dilation of the ventricular cavity, leading to impaired cardiac function and finally to heart failure [48].

Treatment of AMI has focused on avoiding the progression of ischemic heart disease towards heart failure. Cardioprotective therapies such as revascularization by thrombolysis, cardiac intervention and bypass surgery have demonstrated to be useful by improving the blood supply and reversing the adverse remodeling process. Pharmacological therapies have yielded promising outcomes by decreasing heart failure-associated mortality. When the remodeling process is advanced and heart failure becomes chronic, mechanical support therapies, such as left ventricular assist devices and cardiac resynchronization therapy, show beneficial outcomes. However, current therapeutic approaches for heart failure are palliative rather than curative, and heart transplantation remains the only curative solution. Furthermore, organ transplantation is only available for some of the most severe cases due to the donor shortage, the complexity of the surgical procedure and immunocompatibility rejection limitations. In this scenario, understanding cardiac development and regeneration mechanisms can lead to progress in the knowledge of cardiac diseases and the establishment of new therapeutic approaches classified as either cell-based or cell-free. Most studies have focused on the use of stem cells to treat the damaged cardiac tissue.

In the majority of the clinical trials conducted in patients with AMI and chronic ischemic heart failure during the last two decades, intracoronary delivery of bone marrow stem cells has been used, including bone marrow-derived hematopoietic stem cells, endothelial progenitor cells or mesenchymal stem cells. Other first-generation cell types included in clinical trials are skeletal myoblasts or adipose tissue-derived stem cells. Unfortunately, although in most of these studies patient safety was demonstrated, the results failed to meet the expectations of achieving substantial long-term functional benefit [49,50,51,52,53,54]. Second-generation cell types include cells with higher cardiomyogenic potential such as lineage-guided cardiopoietic cells, adult CPCs and pluripotent stem cells (PSC) derived cells [44,55,56,57,58,59,60,61].

### 1.3. The Current Progresses and Challenges to Cell Therapy For Heart Diseases

The lack of oxygen and nutrients caused after AMI leads to acidosis and alterations of ionic homeostasis, generating disturbances in the mitochondrial electron transport chain, preventing the complete reduction of oxygen and giving rise to reactive oxygen species (ROS), and reactive nitrogen species (RNS) [62]. The accumulation of these free radicals generates oxidative stress, which causes apoptosis and necrosis. Cell death, tissue stretching and pro-inflammatory cytokines induce local inflammation orchestrated by bone marrow-derived cells [62]. Inflammation produces cardiac fibroblast activation and mobilization to the injury site, where different components of the extracellular matrix are secreted to repair the damage [63,64]. This stressful environment produced by inflammation and persistent collagen secretion leads to scar formation, where low concentrations of oxygen and glucose are available due to low vascularization, which generates an extremely adverse postinfarction environment.

Thus, although CMs and CPCs obtained through reprogramming methods could be potentially considered ideal cell therapy products, the major problem of cardiac cell therapy still persists, namely, the poor cell retention and low survival of the implanted cells. This is followed by death of most of the remaining cells in the hours and days after transplant, regardless of the cell type used [49,50,51,52,53,54,65]. In fact, any beneficial effect in the restoration of cardiac tissue is attributed to the release of beneficial paracrine molecules by the implanted exogenous cells [55,56,57,58,59,66]. Strikingly, a high degree of cell engraftment and functional recovery in infarcted hearts of non-human primates was recently demonstrated using human PSC-derived CMs. Nonetheless, 750 million cells per subject were required [67].

Thus, the progress of regenerative medicine in the cardiac field will be conditioned by finding new approaches able to retain the cells in the implanted region and increase their survival, and ideally, their engraftment and regenerative potential. Here, recent studies into cell-preconditioning are briefly described.

#### 1.3.1. Cell Pre-Treatments

Subjecting the cells to stressful environmental factors is one of the possible strategies. In recent years, various reports have shown that cultivating cells in hypoxia, heat shock, oxidative stress, or glucose deprivation environments, can improve the survival of transplanted cells. Several studies suggest that cultivating cells in hypoxic conditions leads to the better adaptability of transplanted cells in the ischemic environment and improves cell engraftment, survival, and differentiation. Hu et al. have demonstrated a 20-fold higher graft rate when bone marrow mesenchymal stem cells are cultured under hypoxic conditions (0.87 ± 0.22% in hypoxia vs. 0.045 ± 0.010% in normoxia) [68]. In addition, heat-shock pre-treatment of Sca-1 adult CPCs led to reduced apoptosis and fibrosis and improved cardiac function [69]. On the other hand, the lack of nutrients necessary for transplanted cells to carry out their bioenergetic functions, represents another barrier for recellularizing infarcted tissue. In 2017, Moya et al. induced a reduced metabolic activity or quiescence of mesenchymal stem cells by serum deprivation for two days and demonstrated that pre-conditioned cells were able to survive when cultured in intense anoxia and total glucose depletion. Such cells showed enhanced viability one week after implantation in vivo [70].

Moreover, pharmacological agents, growth factors or cytokines have been used in combination with cells to enhance cell therapeutic activity [71,72].

#### 1.3.2. Genetically Modified Cells

A different approach for cell pre-conditioning is based on genetic modification. There are several molecules that have been shown to play a cardioprotective role against hypoxia such as HASF (Hypoxia and Akt induced Stem cell Factor) [73], hexokinase-2 [74], or sulfiredoxin-1 [75]. The overexpression of these molecules in cells for transplantation purposes could be considered useful strategies that need to be explored. Furthermore, different investigations have shown that the overexpression of different microRNAs (miRs) such as miR-133 [76,77], miR-144 [78], miR-24 [79], miR-335 [80], miR-126 [81], or miR-377 [82] can play a cardioprotective role against AMI and/or enhance angiogenesis. Genetic modification has also been used with the aim of trying to achieve greater adhesion of the transplanted cells to the host tissue. Li et al. overexpressed integrin-β1 and achieved a greater survival rate of the transplanted cells one-week post-transplantation [83]. This is consistent with the observation that integrin-β3 is required for the attachment and retention of transplanted cardiospheres [84]. Finally, Lou et al. very recently used genetically modified human iPSCs with αMHC-CDH2 to overexpress CDH2, also known as N-cadherin, in iPSCs-derived CMs [85]. N-cadherin plays a fundamental role in cell-cell interaction within the myocardium [86] and can stimulate anti-apoptotic pathways [87]. The authors observed a two–three fold greater engraftment rate than with non-modified cells, and reduced infarct size one month after cell transplant [85].

#### 1.3.3. Cells Encapsulated in Biomaterials

A different approach to cell pre-conditioning is the use of biomaterials. Biomaterials can enhance cell survival, prevent anoikis, provide protection, and can even be functionalized to improve the therapeutic potential of the transplanted cells.

Cardiac tissue engineering is a relatively new research field that emerged in response to some of the above-mentioned pitfalls and hurdles encountered in regenerative medicine. Its main goal is to mimic the structure and function of cardiac muscle. However, this becomes particularly challenging when accurate information on the real cardiac architecture is still not clearly defined [88]. Nevertheless, cardiac tissue engineering can reflect more faithfully cardiac tissue than monolayer cell cultures, allowing the creation of in vitro cardiac organoids which could be useful for disease modeling and drug testing [89]. However, it is too early to evaluate the functional benefit of the clinical application of these in vitro engineered cardiac tissues since very few studies are currently being conducted, with only two using iPSC-derived cardiac cells [61,90]. One could venture to predict that cardiac engineered tissues implanted pericardially could improve outcomes compared to previous clinical trials where individual cells were injected and mostly lost, since these scaffolds can prolong the survival and the paracrine action of transplanted cells. However, it becomes difficult to imagine how these engineered cardiac tissues are going to be able to regenerate or functionally replace the damaged tissue.

A different approach is the delivery of the cells encapsulated or embedded in injectable hydrogels. Matrigel and other hydrogels have shown to improve cell survival and cell retention in cardiac tissue [71,91]. Furthermore, different oxygen- or glucose-releasing scaffolds have been developed [92]. Interestingly, it was recently shown that a novel enzyme-controlled glucose starch hydrogel improved the survival of MSC in vitro and in vivo for up to 14 days [93]. Some of these and other pre-conditioning methods have recently been reviewed elsewhere (Lemcke et al. [72], Salazar-Noratto et al. [92], Sart et al. [94], and Abdelwahid et al. [95]).

## 2. Direct Reprogramming for Heart Regeneration

Cell reprogramming has emerged as an attractive approach to obtain de novo cardiac lineages (including CMs and CPCs) for stem cell therapy, disease modeling, drug screening, and developmental biology [96,97,98]. These cells have several advantages over first-generation cell types since CMs and CPCs obtained through cell reprogramming could be expandable or scalable, immunocompatible and integrate and synchronize with the rest of the host myocardium. These cardiac lineages can be obtained from adult somatic cells through different cell reprogramming methods: (1) the establishment of induced pluripotent stem cells (iPSCs) by the overexpression of Oct4, Sox2, Klf4, and cMyc (OSKM) [99] and their subsequent differentiation towards cardiac lineages, or (2) the direct reprogramming into specific cardiac lineages such as induced CMs (iCMs) or iCPCs (iCPCs), bypassing the pluripotent stage, and therefore, avoiding the tumorigenic risk associated with PSCs. Interestingly, CPCs obtained through reprogramming methods are analogous to embryonic cardiogenic mesoderm progenitors (FHF or SHF), have similar gene and protein expression profiles and have the potential to differentiate into the main cell lineages that form the myocardium [7,9,100,101,102,103,104].

In order to establish optimal cardiac reprogramming cocktails, most studies have been performed in vitro from fibroblasts of different origin. However, as described below, there are some reports in which the direct cardiac conversion has been achieved in vivo. One of the major advantages of using an in vivo approach is that direct cardiac reprogramming targets endogenous cells of infarcted heart tissue, thus circumventing the obstacles associated with cell therapy.

In this review, we will thoroughly describe the methods that have been used to obtain iCMs and iCPCs, from the first discoveries until now, and we will discuss the current challenges and gaps that direct cardiac reprogramming faces.

### 2.1. Direct Cardiac Reprogramming In Vitro

#### 2.1.1. Direct Reprogramming into Mouse iCMs

First Discovery: the GMT cocktail

The first described combination of factors for direct cardiac reprogramming of mouse fibroblasts into iCMs was reported by Ieda et al. in 2010 [105]. They carried out a microarray and selected 14 transcription factors overexpressed in mouse embryonic CMs compared to embryonic cardiac fibroblasts. After sequential experiments of single factor withdrawal, they determined that the combination of three transcription factors, GATA4, MEF2C, and TBX5 (GMT), was sufficient and capable of transdifferentiating mouse postnatal cardiac and tail-tip fibroblasts into iCMs using a retroviral delivery system, without passing through a progenitor/stem cell stage. They used αMHC-GFP reporter fibroblasts to detect mature CMs. Although they observed 17% of GFP+ cells, only 30% of these cells coexpressed cTnT. Well-defined sarcomere protein organization and calcium oscillations were found but beating was observed only in 0.01–0.1% of transduced cardiac fibroblasts, and no spontaneous contractions were observed in iCMs derived from tail-tip fibroblasts [105]. Subsequent studies concluded that GMT alone is insufficient to directly reprogram murine adult fibroblast into mature CMs [106].

In order to increase the reprogramming efficiency into iCMs, most studies have focused on the inclusion of additional transcription factors, miRs, small molecules, growth factors, or shRNAs to the original GMT cocktail.

Modifications to the GMT cocktail

Different strategies have been used to improve reprogramming efficiency:

##### Stoichiometric Optimization of GMT Factors

In 2015, Wang et al. generated polycistronic retroviral vectors with GMT factors, separated by identical 2A sequences, and infected them in mouse neonatal cardiac fibroblasts. As splicing order is related to protein expression level, they tested all possible combinations of GMT factor orders and determined that when MEF2C protein levels were higher than GATA4 and TBX5 levels, reprogramming efficiency significantly improved. Moreover, MGT vector achieved a 10-fold increase in mature beating iCMs loci formation compared to GMT in separate vectors, demonstrating the importance of GMT factor stoichiometry in reprogramming efficiency and iCMs maturity [107].

##### Inclusion of Additional Transcription Factors: The Relevance of Hand2 Transcription Factor

After this first description of the GMT cocktail, many groups focused their endeavors on testing new factors to improve cardiac reprogramming efficiency and iCMs maturity. Song et al. introduced a fourth transcription factor, HAND2, to the GMT cocktail (GHMT) [108]. They transduced adult cardiac and tail-tip fibroblasts derived from αMHC-GFP mice with GHMT factors using retroviral vectors and obtained 9.2% and 6.8% of double positive αMHC+/TnT+ iCMs derived from tail-tip and cardiac fibroblasts, respectively. Remarkably, spontaneously beating iCMs were found. These results showed a higher reprogramming efficiency of GHMT combinations compared to GMT alone. The authors reported that iCMs showed intense immunostaining of α-actinin and cTnT and well-organized sarcomeres at day 30 and described three iCMs subtypes (atrial, ventricular, and pacemaker) according to different spectra of cardiac reprogramming [108].

To improve the reprogramming efficiency of GHMT, Umei et al. generated a single-construct-based polycistronic doxycycline (dox) inducible lentiviral vector which expressed both the reverse tetracycline transactivator (rtTA) and the tetracycline response element (TRE) controlled GMT (pDox-GMT), circumventing the limitations of using the conventional dox-inducible systems that require co-transduction of two vectors, one encoding rtTA and another encoding the TRE-controlled transgene. The authors described that infection of cells with pDox-GMT system increased cardiac reprogramming efficiency by three-fold compared to co-transduction with pLVX-rtTA and pLVX-GMT encoding lentiviral vectors. Expression of multiple cardiac genes, sarcomeric structures, and spontaneous contractions were detected. The authors also reported that the reprogramming efficiency in cells co-transduced with pDox-Hand2 and the polycistronic retroviral vector pMX-GMT was three-fold higher than in cells transduced with pMX-GMT alone, and determined that Hand2 was required during the first two weeks of cardiac reprogramming to inhibit cell cycle-promoting genes and enhanced cardiac reprogramming [109].

To enhance transcriptional activity of GHMT, Hirai et al. fused each factor of GHMT to the *MyoD* transactivation domain and overexpressed them in mouse embryonic and neonatal tail-tip fibroblasts using retroviral vectors. They observed that the fusion of the *MyoD* transactivation domain to MEF2C factor produced beating iCMs 15-fold more efficiently than fibroblasts transduced with wild-type MEF2C [110].

Addis et al. included HAND2 and NKX2.5 factors in the GMT cocktail GHMT/NKX2.5 and overexpressed them using dox-inducible lentiviral vectors in mouse embryonic fibroblasts and observed that the efficiency of reprogramming improved up to 50-fold compared to GMT alone. They developed a cTnT-GCaMP5 calcium indicator reporter to detect functional iCMs and observed that 1.6% of transduced fibroblasts were cTnT-GCaMP5+ and expressed CM-specific genes. Calcium oscillations and spontaneous beating persisted for weeks after the inactivation of factors [111].

Christoforou et al. described that the forced expression of GMT with MYOCD and SRF, or GMT with MYOCD, SRF, MESP1, and the BAF chromatin remodeling protein subunit BAF60C, using dox-inducible lentiviral vectors, increased the reprogramming efficiency of mouse embryonic fibroblasts into iCMs compared to GMT alone. They observed sarcomeric protein expression and calcium transients, but no action potentials or contractility were detected [112].

Recently, Zhang et al. described that ensuring the expression co-expression of four GHMT factors in individual fibroblasts, markedly improved direct reprogramming efficiency. First, they infected embryonic fibroblasts with retroviral bi- or tri-cistronic vectors encoding reprogramming factors tagged with different fluorescent proteins (M-T-mCherry plus G-tagBFP or plus G-H-tagBFP). The authors reported that 70–80% of cells expressing GMT or GHMT factors were positive for Titin-eGFP, the third myofilament of cardiac muscle, and α-actinin. However, GHMT expressing cells presented a significantly greater number of spontaneous calcium oscillations and an eight-fold increase in the number of spontaneously beating iCMs compared to GMT expressing cells, demonstrating that Hand2 is a key factor to obtain functional iCMs [113]. Shortly after, the same group demonstrated that the control of stoichiometry of GHMT factors could improve the reprogramming of fibroblasts into contractile iCMs. They generated four retroviral polycistronic vectors encoding GHMT, placing Hand2 in four different positions, and reported that vector with M-G-T-H order of splicing, which induced lower Hand2 protein levels and higher Mef2c protein levels, enhanced direct reprogramming of mouse embryonic fibroblasts into functional iCMs. The authors observed a four- to five-fold increase in sarcomere organization in cells infected with MGTH vector compared to MGT or other constructs, as well as a ~6-fold and ~2.5-fold increase in number of beating iCMs in comparison with MGT and MGHT transduced cells, respectively [114].

##### Addition of miRs

Muraoka et al. demonstrated that the addition of miR-133 (GMT/miR-133) or MESP1 and MYOCD (GMT/MESP1/MYOCD) encoded in lentiviral vectors, improved cardiac reprogramming efficiency of mouse embryonic and adult fibroblasts into iCMs compared to GMT alone, by *Snai1* inhibition and suppression of fibroblast signatures. They observed a seven-fold increase of beating iCMs, and a shorter reprogramming period was required compared to GMT alone [115].

##### Regulation of Signaling Pathways

Many approaches in direct reprogramming have focused on the regulation of signaling pathways or genes that promote the inhibition of fibroblast signatures or enhance cardiac fate. The TGFβ pathway is one of the main signaling pathways active in fibroblasts. Ifkovits et al. reported that the addition of SB431542, a TGFβ inhibitor, to the GHMT/NKX2.5 cocktail produced a five-fold increase in iCMs generation [116]. Regulation of other signaling pathways, such as WNT, Rho-associated kinase (ROCK), JAK/STAT, NOTCH, and AKT in combination with transcription factors and miRs have also shown to improve direct cardiac reprogramming efficiency [112,117,118,119,120,121,122]. Interestingly, Muraoka et al. revealed that the use of diclofenac, a non-steroidal anti-inflammatory drug, in combination with GMT or GHMT cocktails markedly improved cardiac reprogramming, by inhibiting of cyclooxygenase-2, Prostaglandin E2/prostaglandin E receptor 4, cyclic AMP/protein kinase A, and interleukin 1β signaling, and by avoiding inflammation and fibroblast gene program. A three- to four-fold increase in cardiac reprogramming efficiency was detected and a marked enhancement in calcium transients and spontaneous beating compared to GHMT. This effect was observed using postnatal and adult tail-tip fibroblasts but not embryonic fibroblasts, demonstrating that the effect was specific for more mature fibroblasts [123]. On the other hand, in most reprogramming methods, fetal serum has been added to the reprogramming culture medium. However, it is known that serum is composed of chemically undefined constituents and varies between batches so it could interfere and affect reprogramming efficiency. To standardize culture conditions, Yamakawa et al. reported that the addition of FGF2, FGF10, and VEGF (FFV) to a serum-free medium greatly improved direct cardiac reprogramming efficiency. They used mouse embryonic and postnatal tail-tip fibroblasts and overexpressed GMT, GHMT or GMT/MESP1/MYOCD factors, observing a 100-fold increase in beating iCMs compared to serum-based medium, especially when FFV was added at a late stage of reprograming in combination with Wnt inhibitor IWP4 [121].

##### Inhibition of Epigenetic Barriers

Epigenetic modulation has been studied to improve cardiac reprogramming efficiency through inhibition of epigenetic barriers. Zhou et al. focused on the *Bmi1* gene, which was reported to be one of the main epigenetic barriers to cardiac reprogramming, and its inhibition led to an important increase in reprogramming efficiency of mouse fibroblasts into iCMs. To downregulate *Bmi1* in mouse fibroblasts, the authors transduced a pool of *Bmi1* shRNAs, and then these fibroblasts were transduced with different combinations of reprogramming factors (GMT, GHMT). This resulted in a remarkable increase in the number of double positive αMHC+/cTnT+ and beating iCMs 4 weeks after transduction. The authors concluded that depletion of *Bmi1* could substitute GATA4 during iCM reprogramming [124].

Other cocktails different from GMT

MYOCD-MT combination. Protze et al. overexpressed MYOCD, instead of GATA4, together with MEF2C and TBX5 factors in mouse embryonic and neonatal cardiac fibroblasts using a lentiviral delivery system. They reported that the forced expression of these three factors induced the upregulation of a wide range of cardiac genes. Although they obtained 12% of cTnT+ cells, of which only 2% were double positive for cTnT and αMHC, and generated sodium and potassium currents and rare action potentials, no spontaneous beating was observed [125].

miR combo. miRs have been used to enhance direct reprogramming alone or in combination with transcription factors. Jayawardena et al. reported that a transient non-viral transfection system of an miR combo including miR-1, miR-133, miR-208, and miR-499a, (cardiac- and muscle-specific miRs), was able to reprogram αMHC-CFP reporter mice derived mouse cardiac fibroblasts into iCMs in vitro and in vivo [117,126]. Moreover, the addition of JAK inhibitor I improved the efficiency of reprogramming up to 10-fold, obtaining about 28% of CFP+ cells [117].

Chemical cocktail. Fu et al. achieved direct cardiac reprogramming of mouse fibroblasts into iCMs using only chemical cocktails. They described a combination of 6 molecule cocktail called CRFVPT (C, CHIR99021; R, RepSox; F, Forskolin; V, VPA; P, Parnate; T, TTNPB) able to reprogram mouse embryonic and neonatal tail-tip fibroblasts into iCMs [127]. This cocktail, together with DZnep, was firstly described to generate iPSC from mouse embryonic fibroblasts [128]. The authors described that the CRFVPT cocktail converted mouse fibroblasts into iCMs passing through a cardiac precursor stage. About 14.5% of cells were α-actinin+ and 9% of cells were αMHC+ on day 24 after induction. Striations, calcium transients, action potential, and spontaneous beating in the generated iCMs were detected [127].

#### 2.1.2. Direct Reprogramming Into Human iCMs

The initial GMT reprograming cocktail used in mouse cells was demonstrated to be insufficient to convert human fibroblasts into iCMs and the inclusion of additional factors was required.

Modifications to the GMT cocktail

In 2013, Wada et al. revealed that the overexpression of GMT/MESP1/MYOCD in human cardiac and dermal fibroblasts was able to generate iCMs [129]. The GMT/MESP1/MYOCD cocktail induced the expression of a broad range of cardiac-specific proteins in human cardiac and dermal fibroblasts, obtaining up to 5% of iCMs expressed α-actinin and cTnT. Calcium oscillations and action potentials were detected in human iCMs. They also observed contractions of iCMs when they were co-cultured with mouse CMs, but no spontaneous beating iCMs was observed when they were cultured alone [129]. Fu et al. described that the combination of GMT factors plus ESRGG, MESP1, MYOCD, and ZFPM2 using retroviral vectors was able to reprogram human fibroblasts into iCMs. They used αMHC-mCherry hESC-derived fibroblasts, as well as human dermal and cardiac fibroblasts. This combination of factors induced the expression of αMHC (18,1%) and cTnT and the authors were able to observe sarcomere formation, calcium transients, and action potentials. Again, no spontaneous beating was observed [130].

Muraoka et al. described that a GMT, MYOCD, MESP1, and miR-133 combination was sufficient to reprogram human cardiac fibroblasts into iCMs [115]. This cocktail induced the expression of cardiac genes, achieving 23% to 27% of cTnT+ cells of the transfected fibroblasts. However, no spontaneous contractions were detected. Singh et al. used lentiviral vectors encoding GMT, HAND2, MYOCD, or miR-590 and infected rat, porcine, and human cardiac fibroblasts. The addition of HAND2 plus MYOCD, or miR-590 alone, to the GMT induced cTNT expression in approximately 5% of porcine and human fibroblasts and increased the expression of cardiac genes. When porcine iCMs were cocultured with mouse CMs, contractions were detected, but no contractile activity was observed in iCMs obtained from human fibroblasts [131].

Christoforou et al. described that the overexpression of GMT factors plus MYOCD and NKX2.5 using dox-inducible lentiviral vectors, in combination with miR-1 and miR-133 were able to directly convert human dermal fibroblasts into iCMs. They determined that miRs delivery followed by the induction with GMT/MYOCD/NKX2.5 for 2 weeks achieved 3.8 ± 0.8% of cTnT+ cells compared to 0.21 ± 0.04% of cTnT+ cells in condition without miRs, and several cardiac-specific genes were upregulated. Although authors detected calcium transients, they observed immature cytoskeletal organization of both α-actinin and cTnT, and no contractile activity was detected. Additionally, they reported that the addition of JAK1 and GSK3β inhibitors or addition of NRG1 significantly enhanced the efficiency of direct reprogramming [132].

Other cocktails different from GMT

Several authors have added miRs to transcription factor cocktails to enhance the cardiac reprogramming of human fibroblasts. Nam et al. described that a combination of GATA4, HAND2, TBX5, and MYOCD transcription factors together with miR-1 and miR-133 was able to transdifferentiate human neonatal foreskin fibroblasts and adult cardiac and dermal fibroblasts into iCMs using retroviral vectors. They used the cTnT-GCaMP5 reporter. Although they observed that 35% of transfected neonatal foreskin fibroblasts expressed cTnT and calcium transients were detected, scarce contractions were observed in these iCMs. However, no beating cells were detected from transduced adult human fibroblasts [133].

Cao et al. described nine compounds (CHIR99021, A83-01, BIX01294, AS8351, SC1, Y27632, OAC2, SU16F, and JNJ10198409) that were able to convert human neonatal dermal fibroblasts into functional αMHC-GFP+ iCMs. These iCMs could form beating clusters and expressed specific cardiac proteins. Remarkably, although the reprogramming efficiency was low using this procedure, the iCMs obtained were homogeneous and functional [134].

#### 2.1.3. Direct Reprogramming Into iCPCs

As can be seen, many protocols have been described for the direct reprogramming of fibroblasts into iCMs, even though most of these are based on the initial GMT cocktail. Nevertheless, iCMs have several limitations in cardiac regeneration therapy. First, these cells have no or limited proliferative capacity, which limits cell yield. Furthermore, the myocardium apart from CMs is made up of blood vessels (vascular endothelial cells and smooth muscle cells) and fibroblasts, and therefore, other cell components besides CMs will be required for an appropriate cardiac regeneration. Taking these considerations into account, some groups have focused on obtaining iCPCs, since these cells retain proliferative potential and are able to give rise to all myocardial cell lineages.

First Discovery: the ETS2 and MESP1 combination

In 2012, Islas et al. selected ETS2 and MESP1 transcription factors, based on the knowledge that homologous genes in ascidian *Ciona* (Ci-ets1/2 and Ci-mesp) were key regulators of cardiogenesis, and observed that these factors were capable of transdifferentiating human dermal fibroblasts into iCPCs in vitro [135]. They found that the overexpression of ETS2 alone induced cardiovascular gene expression, such as KDR and PECAM1, but was insufficient to induce complete reprogramming. Additionally, they demonstrated that forced coexpression of MESP1 and ETS2 with lentiviral vectors or with purified proteins, and subsequent addition of Activin A and BMP2 was able to convert human dermal fibroblasts into iCPCs, which were identified by an *NKX2.5* lentiviral reporter system. Nonetheless, these progenitors were not well characterized and spontaneously differentiated into immature CMs and the potential to differentiate into smooth muscle cells or endothelial cells was not demonstrated [135].

Modifications to the GMT cocktail

Pratico et al. generated cKit+ iCPCs by the electroporation of human adult dermal fibroblasts with GHMT mRNAs. These iCPCs expressed NKX2.5 an ISL1 and differentiated into cTnT+ CMs with a very low efficiency and were non-contractile [136].

In 2015, Li et al. transduced human dermal fibroblasts with GHMT proteins modified with QQ-reagent, a non-viral delivery system, obtaining iCPCs [137]. They also added BMP4 and Activin A for cardiac induction and bFGF to maintain the CPC stage and observed a downregulation of fibroblast specific genes and an upregulation of CPC markers. Moreover, they demonstrated that iCPCs were able to differentiate by WNT inhibition into three principal cardiac lineages: smooth muscle cells (2–5%), endothelial cells (15–20%), and mainly CMs (>70%), which started beating after 20 days of differentiation. Finally, they observed improvement in heart function and tissue remodeling when iCPCs were injected in a rat model of AMI [137]. Recently, the same group directly reprogrammed human foreskin fibroblasts into iCPCs using a death Cas9 (dCas9)-based transcription activation system [138]. This novel system consisted of the fusion of a deactivated form of Cas9 (dCas9) with transactivation domains of VP64 and p300, generating a precise synthetic transcription activator. The authors infected dCas9 expressing fibroblasts with lentiviral vectors encoding specific sgRNAs directed to the promoter regions of GHMT and highlighted that the transactivation of an additional factor, MEIS1, could improve reprogramming efficiency. They observed an overexpression of EOMES and MESP1 genes at an early reprogramming stage and CPC-related markers such as NKX2.5 and ISL1 at days 7–10 post-induction. They observed differentiation into three main cardiac lineages 4 weeks post-induction. They detected expression of CM markers: cTNT (~6% with GHMT or 8.75% when MEIS1 was also transactivated), expression of α-actinin, sarcomere striations, as well as smooth muscle and endothelial marker expression. Unfortunately, the CMs obtained were not functional and no beating was observed [138].

Other cocktails different from GMT: expandable mouse iCPCs

There are only two groups that have achieved expandable multipotent iCPCs using mouse cells [104,139].

In 2016, Lalit et al. successfully converted adult mouse fibroblasts into iCPCs using five factors, including MESP1, TBX5, GATA4, and NKX2.5 transcription factors and BAF60c [104]. To detect reprogrammed iCPCs and achieve a dox-inducible transgene expression, an *Nkx2.5*-EYFP cardiac reporter mouse model was crossed with a transgenic mouse expressing the reverse tetracycline transactivator. Then, based on their implication in early cardiac development, 22 factors were selected and cloned individually in a lentiviral vector with the tetracycline response element. Although they could observe EYFP+ cells after transduction with all the factors, the reprogramming medium (fibroblast medium supplemented with dox) was insufficient for long-term maintenance of proliferative EYFP+ cells. To achieve long termed maintenance, BIO, an activator of canonical WNT signaling pathway, and LIF, an activator of JAK/STAT, was supplemented to the reprogramming medium, which enabled iCPCs expansion and maintenance in the absence of dox for over 30 passages. Next, they selected the minimal combination of factors able to reprogram fibroblasts into expandable iCPCs: MESP1, TBX5, GATA4, NKX2.5, and BAF60C. They demonstrated that the iCPCs generated were able to differentiate into three main cardiac lineages in vitro. These iCPCs principally differentiated into CMs (80% to 90%), to a lesser extent into smooth muscle cells (5% to 10%) and scarcely generated endothelial cells (1% to 5%). The iCPC-derived CMs did not beat spontaneously, and only when cocultured with mouse PSC-derived CMs did 5% to 10% of the iCPC-derived CMs start beating. Nonetheless, the authors demonstrated the capacity of iCPCs to give rise to myocardial cell lineages when injected into the cardiac crescent of mouse embryos and in adult mouse hearts after AMI [104].

Simultaneously, Zhang et al. described culture conditions able to highly expand iCPCs obtained from mouse embryonic and tail-tip fibroblasts using the cell activation and signaling-directed (CASD) conversion approach [139]. To capture and expand iCPCs, they tested modulators of different signaling pathways, such as SMADS, FGF, VEGF, PDGF, WNT, and NOTCH. Finally, they described BACS, a combination of molecules that contains BMP4, Activin A, CHIR99021 (GSK3β inhibitor), and SU5402 (FGF, VEGF, and PDGF inhibitor), which repressed cardiovascular differentiation and supported iCPCs self-renewal and expansion. iCPCs were purified using FLK1 and PDGFRα surface markers. The authors demonstrated that FLK1+/PDGFRα+ cells expressed CPC-related markers such as GATA4, MEF2C, ISL1, and NKX2.5, suggesting that these cells were committed to a cardiovascular fate and reported that in BACS conditions they were expandable for more than 18 passages. Additionally, these iCPCs were able to differentiate into the main cardiac lineages (CMs, smooth muscle and endothelial cells) in vitro using specific differentiation conditions. To obtain CMs, they added the WNT inhibitor IWP2 and found that 35% of cells were cTnT+ and expressed CM-specific markers. The number of spontaneously contracting cells gradually increased until day 10 of differentiation and iCMs expressed a well-organized sarcomeric structure, calcium transients, and action potentials. They also reported that about 90% of iCPCs expressed CD31 and mature endothelial cell features under endothelial cell differentiation conditions for 10 days, and more than 98% of iCPCs expressed smooth muscle cell-specific markers after 10 days of smooth muscle differentiation. When iCPCs were transplanted into mouse models of AMI, improved cardiac function and reduced adverse remodeling was observed [139]. However, it needs to be mentioned that the CASD approach generates a pluripotent intermediate state [140,141], and therefore, it cannot be considered a direct reprogramming approach.

### 2.2. Direct Cardiac Reprogramming In Vivo

On the hypothesis that heart environment (extracellular matrix, growth factors, cytokines, electromechanical stimulation, etc.) may favor direct cardiac reprogramming in vivo, several studies have been carried out.

The GMT cocktail

In 2012, Qian et al. revealed that the injection of a retrovirus (ReV) encoding individual GMT factors in peri-infarcted area of mouse hearts was able to reprogram resident cardiac fibroblasts into iCMs [142]. They used two independent fibroblast reporter mice (Periostin-Cre: R26R-LacZ and Fsp1-Cre: R26R-LacZ) in order to verify that iCMs were derived from resident fibroblasts. In addition, they used an *αMHC* transgenic mouse to demonstrate that iCMs were not formed by fusion of endogenous CMs. To induce resident cardiac fibroblast proliferation and improve reprogramming efficiency, they injected thymosin β4, a fibroblast-activating peptide, in combination with GMT factors. They detected α-actinin expression in approximately 12% of transduced cells, and 50% of these had mature CM features, according to structure, electrophysiology, and contractility. They also observed an improvement in heart structure and function 8 to 12 weeks post injection [142].

Since the heterogenicity of factor delivery generated by using multiple viral vectors encoding individual factors could consequently decrease reprogramming efficiency, several groups have focused on the use of polycistronic vectors encoding different splicing orders of GMT factors. Inagawa et al. showed that the injection of a 2A-polycistronic retroviral vector encoding TMG factors in infarcted mouse hearts increased two-fold higher the number of mature iCMs compared to the three single vectors [143]. Ma et al. injected PT2A-polycistronic retroviral MGT vector into mouse infarcted hearts, which increased the number of iCMs generated, but not their maturity. Moreover, they observed an improvement in heart function and structure compared to single retroviral vectors [144]. Mathison et al. used TE2A-policystronic lentiviral vectors encoding GTM factors and injected them in rat models of chronic MI. First, the authors injected adenoviral vectors encoding VEGF in rat hearts immediately after infarction, to enhance vascularization, and GMT lentivirus (LeV) were injected after 3 weeks. They observed an improvement in cardiac remodeling and function as well as a decrease in the number of myofibroblasts compared to monocystronic vectors [145,146].

Mutagenesis is one of the major concerns regarding integrative viral vectors. Recently, some groups have used non-integrative viral vectors, such as adenovirus (AV) or Sendai virus (SeV) vectors. Mathison et al. demonstrated that AV and LeV induced equivalent expression levels of GMT factors and had a similar transdifferentiation capacity of rat fibroblasts into iCMs in vitro [147]. When AV encoding GMT factors were injected into the infarcted rat myocardium, an increase in cells expressing βMHC and improved ejection fraction compared to empty AV was observed [147]. Miyamoto et al. generated SeV polycistronic GMT vectors and injected them into mouse hearts after AMI, achieving direct reprogramming of resident CFs into iCMs in vivo with a greater efficiency compared to retroviral polycistronic vectors [148]. They used Tcf21iCre/R26-tdTomato mice and observed that 1.5% of Tomato+ cells were cTnT+, and 20% of these double positive cells presented cross striations 1 week after injection. They also detected an improvement in cardiac function and a reduction in fibrosis [148].

Finally, Chang et al. reported that polyethylimine (PEI) conjugated cationic gold nanoparticles (AuNPs) loaded with GMT operate as nanocarriers for cardiac direct reprogramming in vitro and in vivo. Functional iCMs were obtained from mouse embryonic fibroblasts using AuNP/GMT/PEI in vitro. Remarkably, AuNP/GMT/PEI nanocomplexes produced efficient in vivo conversion of resident fibroblasts into iCMs when injected in mouse models of AMI, resulting in reduced infarct size and fibrosis, and improved cardiac function. Importantly, they determined that this approach did not generate DNA integrations and had low toxicity [149].

Modifications to the GMT cocktail

In 2012, Song et al., concomitantly with Qian et al. [142], reported that the injection of GHMT factors, encoded individually in retroviral vectors, could trigger the conversion of endogenous cardiac fibroblasts into iCMs in mouse models of AMI [108]. They used both Fsp1-Cre/Rosa26-LacZ and Tcf21iCre/R26-tdTomato mice to trace reprogrammed non-CMs cells and observed that about 6.5% of CMs in the injured area displayed β-gal activity. These β-gal expressing cells also expressed cTnT and showed striations 3 weeks after viral transduction. Moreover, they detected the presence of calcium transients, action potentials, and contractility. More importantly, GHMT-infected hearts showed a pronounced reduction of scar size and increased muscle tissue and improved heart function 12 weeks post infarction [108].

Other cocktails different from GMT

Jayawardena et al. reported that an miR combo (miR-1, miR-133, miR-208, and miR-499a) was able to reprogram resident cardiac fibroblasts into CMs in vivo when injected in infarcted mouse hearts. They used LeV to overexpress the miR combo and used Fsp1-Cre/tdTomato mice. They observed that 12% of CMs of the peri-infarcted zone were tdTomato+/cTnT+ and organized sarcomeres, action potentials, and contractility were detected. Additionally, improvement in several cardiac function parameters was reported [117,126].

The combination of reprogramming factors with chemical inhibitors has also been tested in vivo to enhance reprogramming efficiency and CM maturity. Mohamed et al. injected GMT factors, encoded in retroviral vectors, in combination with repeated intraperitoneal administration of SB431542, a TGFβ inhibitor, and XAV939, a WNT signaling inhibitor, in an AMI mouse model. ROSA-YFP/periostin-Cre mice were used, and a five-fold increase in YFP+ iCMs was detected in mice treated with GMT plus inhibitors compared to GMT treated mice and iCMs were more mature. Moreover, the addition of both small molecules led to attenuated remodeling and improved cardiac function [122].

Huang et al., achieved direct conversion of mouse fibroblasts in vivo using only a chemical cocktail. They added Rolipram to a previously described CRFVPT chemical cocktail [127] (CRFVPTM: C, CHIR99021; R, RepSox; F, Forskolin; V, VPA; P, Parnate; T, TTNPB; M, Rolipram) and this combination was able to induce the generation of iCMs from cardiac fibroblasts in hearts of healthy adult mice. CRFTM were administered orally and VP intraperitoneally [150]. They used Fsp1-Cre: R26R^tdTomato^ mice to evaluate the non-CM origin of iCMs. CM-like tdTomato+ cells expressed CM-specific markers, including α-actinin, cTnI, GATA4, and MEF2C, and showed a well-formed sarcomeric structure and action potentials. In infarcted hearts they observed that the CRFVPTM combination was also able to reduce the formation of fibrosis. Intriguingly, this group described that fibroblasts from other organs were not converted into iCMs, which indicates that this cocktail specifically reprograms cardiac fibroblasts [150].

All the aforementioned reprogramming cocktails used for direct cardiac reprogramming are summarized in Table 1.

## 3. Future Directions and Challenges

The increasing knowledge about the gene regulatory networks and signaling pathways that regulate cardiac development and the underlying mechanisms that control cell reprogramming, has led to significant advances in the cardiac reprogramming field in recent years.

Direct cardiac reprogramming offers several advantages over PSCs, since it is a faster process, avoids the pluripotency stage (and therefore tumorigenic risk), and offers the opportunity of direct injection of defined factors to convert directly endogenous fibroblasts into cardiac lineages in the native environment of injured cardiac tissue. In contrast, human CMs and CPCs obtained through the directed differentiation of PSC are more homogenous and show improved functionality over iCMs and iCPCs generated by direct reprogramming approaches. In addition, the current CM differentiation protocols can reach 80–98% efficiency without selection and be effective in multiple PSC lines [151,152]. However, we need to remember that embryoid body-based protocols developed two decades ago achieved a poor differentiation efficiency, as low as 8.1% [153], and that these successful and robust differentiation protocols that are currently highly reproducible in many different laboratories all over the world, were developed after many studies and optimizations of differentiation conditions. This could also be the case of direct cardiac reprogramming technology, in that it is only a matter of time and further investigation.

The enhancement of reprogramming efficiency is a main challenge in direct cardiac reprogramming. In this respect, standardized criteria to measure reprogramming efficiency should be applied in this field. First results demonstrated that establishing reprogramming efficiency cannot rely on the use of reporters such as αMHC after the observations that only part of αMHC+ cells co-expressed other cardiac-related structural proteins such as cTnT or showed contractility, indicating that not all cells that activated the αMHC promoter were completely reprogrammed into iCMs [105].

Most of the direct cardiac reprogramming approaches described have been based on the original GMT cocktail. The identification of additional transcription factors, miRs or small molecules, has allowed reprogramming efficiency to be improved. Remarkably, the addition of Hand2 to GMT cocktail, firstly described by Song et al. [108], demonstrated an increase in the expression of cardiac proteins and more importantly the generation of spontaneously contracting iCMs, in contrast to the original GMT cocktail [106]. Hand2 has been included in many of the in vitro direct cardiac reprogramming approaches using mouse fibroblasts [108,110,111,116,118,119,120,121,124]; in addition, this factor has been used in a reprogramming cocktail for in vivo conversion [108]. Importantly, Hand2 has been incorporated in reprogramming cocktails to convert human fibroblasts into iCMs [131,133] and for the generation of iCPCs [136,137,138]. Recent studies have demonstrated the crucial effect of Hand2 on the functionality of iCMs [109,113] and the importance of the stoichiometric optimization of the reprogramming factors [114].

Nonetheless, as can be noted from the studies presented here, there is still no consensus about which of these GMT or GHMT varieties is best and sufficient for efficient cardiac reprogramming. Epigenetics and signaling pathways activated in the starting fibroblast population, which are different depending on the species and developmental origin, may be the major barriers to cardiac cell fate conversion, together with the complex and unique filament structure that needs to be reconstructed to achieve functional iCMs. In line with this, some studies have used the inhibition of the TGF-β signaling pathway [116,122] or the knockdown of Bmi1 [124], both of which play a fundamental role at the early stages of cardiac reprogramming, in order to increase reprogramming efficiency. On the other hand, the lower cardiac reprogramming efficiency in human cells compared to mice cells, and the requirement of additional factors, is a worrying issue. Therefore, more emphasis should now be placed on the establishment of robust and well-defined protocols to create human iCMs and iCPCs that could be highly reproducible in different laboratories worldwide. Once this protocol is well-established, it should be tested in large animal models before being translated into the clinic. To this end, it might be necessary to identify novel cocktails which may or may not be based on the GMT cocktail. In this respect, in our laboratory, we are currently testing novel reprogramming factors selected using bioinformatics-based approaches in conjunction with GMT and other well-known cardiac transcription factors for direct cardiac reprogramming (unpublished results).

The enhancement of reprogramming efficiency to obtain iCMs is critical to produce sufficient cells in vitro for transplantation. To solve this issue, alternatively, direct reprogramming of fibroblasts into expandable iCPCs can be induced in vitro [104]. These cells could be more malleable and adaptable to the recipient than terminally differentiated iCMs. They retain their proliferative activity and have the potential to differentiate into all the myocardial cell lineages. However, studies of direct reprogramming of fibroblasts into iCPCs are scarce. In addition, it is surprising that in some reports the GHMT cocktail, described previously to be useful for mouse iCMs [108] but not for human iCMs conversion [133], has been used for human iCPCs reprogramming [137,138]. Despite these inconsistencies, interestingly, some studies have demonstrated the long-term expansion of multipotent mouse iCPCs in culture, and the spontaneous differentiation of these iCPCs into three cardiovascular cell types when transplanted in vivo, indicating that injured heart tissue can still provide environmental cues to direct the differentiation of iCPCs [104,139]. Obtaining expandable human iCPCs in culture has still not been achieved. Nonetheless, the retention and integration of transplanted cells, and the correct regeneration and reconstruction of the cardiac tissue by iCPCs or iCMCs, remain a major concern. In this context, cell pre-conditioning with any of the strategies described herein or the development of novel approaches to enhance their survival and therapeutic potential will be fundamental for the progress of regenerative medicine in the cardiac field.

Against this background, in vivo reprogramming may face fewer hurdles for its translation to the clinic than cell injection approaches. The heart microenvironment has been found to be more favorable to cardiac reprogramming than the in vitro culture conditions in terms of efficiency and maturation of mouse iCMs [108,142,144]. In this regard, although human cardiac reprogramming has not demonstrated to be an efficient process in vitro, it may act better in vivo. On the other hand, the direct reprogramming of fibroblasts into iCPCs in vivo remains to be explored and the potential of this strategy for cardiac regeneration requires further evaluation. Despite encouraging results obtained with iCMs, this approach taken in vivo still needs improvements in safety and routes of delivery. AV [147] and SeV [148] mediated reprogramming are suitable alternatives. However, the major handicap of these integration-free vectors is their small packaging capacity which precludes the inclusion of all the required reprogramming factors, which is especially relevant in the case of human cells, and consequently hinders effective delivery. Other safe alternatives would be the use of engineered proteins and synthetic RNAs for in vivo reprogramming, but their short half-life reduces their effectiveness. Alternatively, CRISPR activation system can induce endogenous gene expression and avoid exogenous gene expression [138]. However, the efficient delivery of this system still relies on the use of viral vectors since other alternative transfection methods used to transduce this system (Cas9-gRNA ribonucleoproteins) in vitro such as nucleofection, are not suitable for application in vivo. The use of chemical cocktails may be an ideal option for reprogramming in vivo [150], since small molecules are cost-effective, have a long half-life, constitute an integration-free system, and are non-immunogenic. Other safe and interesting system to deliver genes, proteins, and small molecules could be the use of nanoparticles [149,154].

Finally, direct reprogramming can be used for other applications other than cardiac regeneration, such as disease modeling or drug testing. Multiple diseases have been modeled in vitro using iPSCs, including cardiac diseases [96,97,98]. Direct reprogramming could be considered an interesting approach for modeling aging-related diseases, since it avoids rejuvenation and retains the hallmarks of cellular aging, which on the contrary, might not be useful for recapitulating early development events, crucial for the study of monogenic heart diseases. Although disease models of neurologic illnesses, many of which are associated with age, have been reported [155], cardiac disease models using direct reprogramming approaches have not yet been described.

In this review we describe the published strategies used for the direct reprogramming of fibroblasts into iCMs and iCPCs, in mouse and human cells, and discuss the remaining challenges and hurdles that need to be overcome before these technologies can be applied for the treatment of heart failure (represented in Figure 1). Despite the current limitations, it is clear that direct cardiac reprogramming holds great promise for regenerative therapy. 

## Figures and Tables

**Figure 1 ijms-21-07950-f001:**
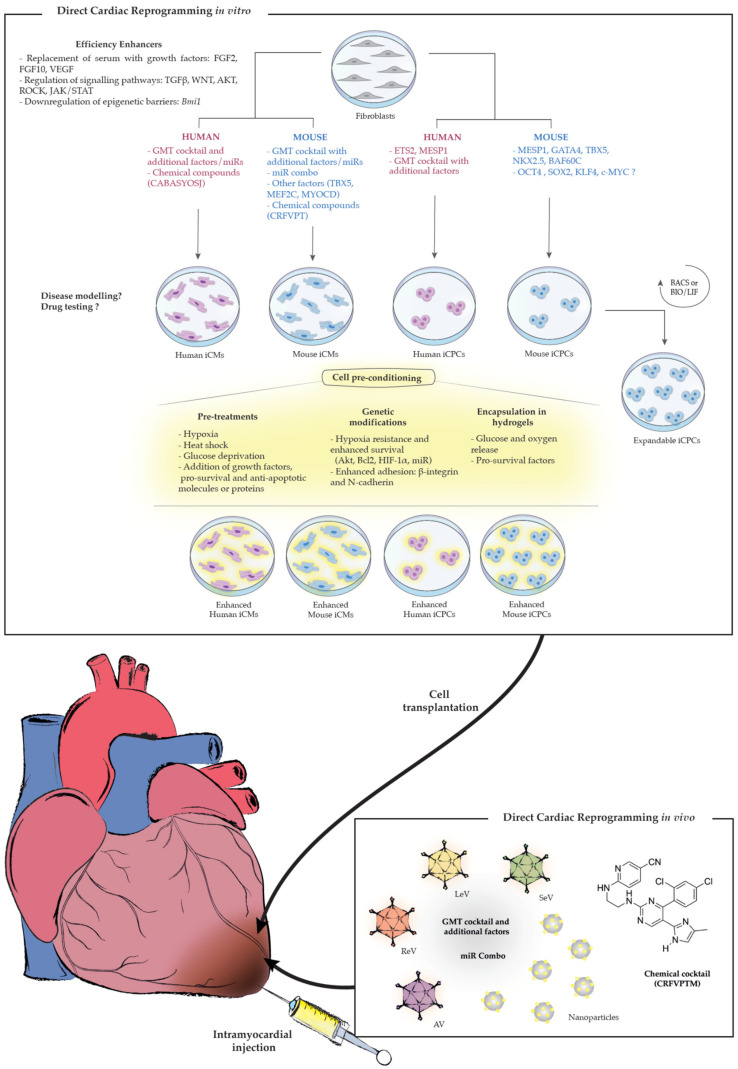
Schematic representation of different approaches used for in vitro and in vivo direct cardiac reprogramming.

**Table 1 ijms-21-07950-t001:** Summary of reprogramming cocktails used for direct cardiac reprogramming. Abbreviations: iCMs: induced cardiomyocytes; CFs: cardiac fibroblasts; TTFs: tail-tip fibroblasts; MEFs: mouse embryonic fibroblasts; hESC: human embryonic stem cell; HDFs: human dermal fibroblasts, HCFs: human cardiac fibroblasts; HFFs: human foreskin fibroblasts; ND: not defined; AP: action potentials; CaT: calcium transients; SB: spontaneous beating; c-B: beating when co-cultured with murine CMs; iCPCs: induced cardiac progenitors; Tri-lineage dif.: tri-lineage differentiation potential; ReV: retrovirus; SeV: Sendai virus vector; AV: adenovirus; LeV: lentivirus; HF: heart function; CS: cardiac tissue structure.

Cell Origin	Reprogramming Cocktails	Efficiency	Functionality	References
**Direct cardiac reprogramming into iCMs in vitro**				
**GMT and modifications to GMT cocktail**				
Mouse	GATA4, MEF2C, TBX5	4-6% αMHC-GFP+/cTnT+ iCMs from CFs	AP,CaT, SB	[105]
	MEF2C, GATA4, TBX5	~10% αMHC-GFP+ and ~4.8% cTnT+ iCMs from CFs	AP,CaT, SB	[107]
	GATA4, MEF2C, TBX5, HAND2	9.2% and 6.8% αMHC+/TnT+ iCMs from TTFs and CFs, respectively	CaT, SB	[108]
	GATA4, MEF2C, TBX5, HAND2	~1.5% cTnT+ in pDox-GMT; 13% cTnT+ in pMX–GMT/pDox–Hand2 iCMs, from MEFs	CaT, SB	[109]
	GATA4, MEF2C, TBX5, HAND2	~70–80% of cells expressing GMT(H) were Titin-eGFP+ or α-actinin+ iCMs from MEFs	CaT, SB	[113]
	MEF2C, GATA4, TBX5, HAND2	~25% Titin-eGFP+/α-actinin+ iCMs from MEFs	CaT, SB	[114]
	GATA4, MYOD-MEF2C, TBX5, HAND2	10-20% cTnT+ iCMs from embryonic head fibroblasts	CaT, SB	[110]
	GATA4, MEF2C, TBX5, HAND2, NKX2.5	1.6% cTnT-GCaMP5+ iCMs from MEFs	CaT, SB	[111]
	GATA4, MEF2C, TBX5, MYOCD, SRF, (MESP1, BAF60C)	2.4% αMHC-GFP+ iCMs from MEFs	CaT, no SB	[112]
	GATA4, MEF2C, TBX5, (miR-133 or MESP1, MYOCD)	9.5% αMHC-GFP+/ cTnT+ and 19.9% α-actinin+ iCMs from MEFs	CaT, SB	[115]
	GATA4, MEF2C, TBX5, HAND2, NKX2.5, SB431542	17% cTnT-GCaMP5+ iCMs from MEFs; 9.27% cTnT-GCaMP5+ iCMs from CFs	CaT, SB	[116]
	GATA4, MEF2C, TBX5, HAND2, DAPT	~38% cTnT+ and ~35% α-actinin+ iCMs from MEFs	CaT, SB	[118]
	GATA4, MEF2C, TBX5, HAND2, miR-1, miR-133, A83-01, Y-27632	60% cTnT+ and 60% α-actinin+ iCMs from MEFs	AP, CaT, SB	[119]
	GATA4, MEF2C, TBX5, HAND2, AKT1	23.3% αMHC-GFP+/cTnT+ iCMs from MEFs; 50% beating iCMs from MEFs at Day 21	CaT, SB	[120]
	GATA4, MEF2C, TBX5, (HAND2 or MESP1, MYOCD), FGF2, FGF10, VEGF	~13% αMHC-GFP+ and ~2% cTnT+ iCMs from MEFs	CaT, SB	[121]
	GATA4, MEF2C, TBX5, SB431542, XAV939	~30% αMHC-GFP+ iCMs from CFs	AP,CaT, SB	[122]
	GATA4, MEF2C, TBX5, HAND2, Diclofenac	~5% cTnT+/ αMHC+ iCMs from postnatal TTFs	CaT, SB	[123]
	GATA4, MEF2C, TBX5, (HAND2), Bmi1 shRNA	22% αMHC+/TnT+ iCMs from CFs	CaT, SB	[124]
Human	GATA4, MEF2C, TBX5, MESP1, MYOCD	5.9% cTnT+ and 5.5% α-actinin+ iCMs from HCFs	AP, CaT, c-B	[129]
	GATA4, MEF2C, TBX5, ESRGG, MESP1, MYOCD, ZFPM2	13% αMHC-mCherry+/cTnT+ iCMs from hESC-derived fibroblasts	AP, CaT, no SB	[130]
	GATA4, MEF2C, TBX5, MESP1, MYOCD, miR-133	27.8% cTnT+ and 8% α-actinin+ iCMs from HCFs	CaT, no SB	[115]
	GATA4, MEF2C, TBX5, MYOCD, NKX2.5, mir-1, miR-133, JAK1i, GSK3βi or NRG	~3.8% cTnT+ iCMs from HDFs	CaT, no SB	[132]
Human, rat, porcine	GATA4, MEF2C, TBX5, (HAND2, MYOCD or miR-590)	~40% αMHC-GFP+ and ~5-6% cTnT+ iCMs from adult HCFs	No SB in human iCMs	[131]
**Other cocktails different from GMT**				
Mouse	TBX5, MEF2C, MYOCD	~11% cTnT+ iCMs from CFs	AP	[125]
	miR-1, miR-133, miR-208, miR-499a, JI1	~28% αMHC-CFP+ iCMs from CFs	AP, CaT, SB	[117]
	CHIR99021, RepSox, Forskolin, VPA, Parnate, TTNPB	14.5% α-actinin+ and 9% α-MHC+ iCMs from MEFs	AP, CaT, SB	[127]
Human	GATA4, HAND2, TBX5, MYOCD, miR-1, miR-133	~35% cTnT+ and ~42% tropomyosin+ iCMs from HFFs	CaT, SB	[133]
	CHIR99021, A83-01, BIX01294, AS8351, SC1, Y27632, OAC2, SU16F, JNJ10198409	7% cTnT+ iCMs from HFFs	AP, CaT, SB	[134]
**Direct reprogramming into iCPCs in vitro**				
Mouse	MESP1, TBX5, GATA4, NKX2.5, BAF60C, BIO, LIF	> 90% Nkx2.5-YFP+, Gata4+ and Irx4+ iCPCs from adult CFs	Expandable; Tri-lineage dif.; *In vivo* in AMI	[104]
	OCT4, SOX2, KLF4, C-MYC, BMP4, Activin A, CHIR99021, SU5402	70% Flk1+/Pdgfrα+ iCPCs from MEFs	Expandable; Tri-lineage dif.; *In vivo* in AMI	[139]
Human	ETS2, MESP1, Activin A, BMP2	9.3% NKX2.5-tdTomato+ iCPCs from HDFs	Not expandable; Unipotent (CM)	[135]
	GATA4, MEF2C, TBX5, HAND2	4.9% c-Kit+ iCPCs from adult HDFs	Not expandable; Unipotent (CM)	[136]
	GATA4, MEF2C, TBX5, HAND2, BMP4, Activin A, bFGF	81% Flk1+ and 83% Isl1+ iCPCs from HDFs	Not expandable; Tri-lineage dif.; *In vivo* in AMI	[137]
	GATA4, MEF2C, TBX5, HAND2	~72% of GATA4+ cells were NKX2.5+; ~85% of HAND2+ cells were ISL1+, from HFFs	Not expandable; Tri-lineage dif.	[138]
**Direct reprogramming into iCMs in vivo**				
**GMT cocktail**				
Mouse	GATA4, MEF2C, TBX5, (ReV vector), Thymosin β4 (intramyocardial)	Periostin-Cre: R26R-lacZ mice: 35% β-Gal+ and α-actinin+ iCMs	Improvement in HF and CS	[142]
	TBX5, MEF2C, GATA4 (ReV vector)	1% α-actinin+ iCMs derived from GMT transduced cells	Improvement in HF and CS	[143]
	MEF2C, GATA4, TBX5 (ReV vector)	Periostin-Cre: R26R-lacZ mice: ~80 β-Gal+/α-actinin+ iCMs per section	Improvement in HF and CS	[144]
	GATA4, MEF2C, TBX5 (SeV vector)	TCF21iCre/R26-tdTomato mice: ∼1.5% tdTomato+/cTnT+ iCMs	Improvement in HF and CS	[148]
	GATA4, MEF2C, TBX5 (Nanoparticles)	*In vitro*: 22% αMHC-eGFP+ iCMs from MEFs; *In vivo*: ND	Improvement in HF and CS	[149]
Rat	GATA4, MEF2C, TBX5 (AV vector)	*In vitro*: ~6.5% cTnT+ iCMs from rat CFs; *In vivo*: ND	Improvement in HF and CS	[147]
	GATA4, TBX5, MEF2C (LeV vector), VEGF (AV vector)	ND	Improvement in HF and CS	[145,146]
**Modifications to GMT cocktail**				
Mouse	GATA4, MEF2C, TBX5, HAND2 (ReV vector)	Fsp1-Cre x R26LacZ mice: ~6.5% β-Gal+ iCMs; TCF21-iCre x R26tdTomato mice: ~2.4% tdTomato+ iCMs	Improvement in HF and CS	[108]
**Other cocktails different from GMT**				
Mouse	miR-1, miR-133, miR-208, miR-499a (LeV vector)	Fsp1-Cre: R26R-tdTomato mice: 12% tdTomato+/cTnT+ iCMs	Improvement in HF and CS	[126]
	GATA4, MEF2C, TBX5 (ReV vector) SB431542, XAV939 (intraperitoneal)	ROSA-YFP/Periostin-Cre mice: 150-200 YFP+/cTnT+ iCMs per section	Improvement in HF and CS	[122]
	CHIR99021, RepSox, Forskolin, TTNPB, Rolipram (oral) VPA, Parnate (intraperitoneal)	Fsp1-Cre: R26RtdTomato: 0.78% tdTomato+/α-actinin+ iCMs	Improvement in HF and CS	[150]

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
