# Peer review of "The Future of Direct Cardiac Reprogramming: Any GMT Cocktail Variety?"

_ijms, 2020, doi:10.3390/ijms21217950_

Round 1

Reviewer 1 Report

In the study, titled "The future of direct cardiac reprogramming: any GMT cocktail variety", Lopez-Muneta L et al., review the current literature regarding direct reprogramming of various cardiac cell types into cardiomyocytes for cardiac repair and regeneration. Authors have done really well in summarizing the current literature on the topic. There are few comments that will increase the impact of the article.

  1. The title is unclear. What do the authors implying with the statement "any GMT cocktail variety". It would be good to rephrase for better clarity.
  2. Authors should provide an illustration or a section that discusses the advantages or limitations of direct reprogramming vs cell-based therapeutics or endogenous CM proliferation response. 
  3. One of the biggest limitations with direct reprogramming is the low efficiency of conversion into CMs. Authors should discuss this in a separate section together with discussion of the current literature that targets enhancement of CM generation. A table representation may be helpful as well.
  4. First 2 sections, "Cardiogenesis" and "Regenerative medicine" can be shortened.
  5. It may helpful if the authors could provide a table showing the impact of every direct reprogramming approach on cellular functions such as survival, proliferation and cardiac commitment.

Author Response

In the study, titled "The future of direct cardiac reprogramming: any GMT cocktail variety", Lopez-Muneta L. et al., review the current literature regarding direct reprogramming of various cardiac cell types into cardiomyocytes for cardiac repair and regeneration. Authors have done really well in summarizing the current literature on the topic. There are few comments that will increase the impact of the article.

1. The title is unclear. What do the authors implying with the statement "any GMT cocktail variety". It would be good to rephrase for better clarity. After much thought and consideration, all the authors agree and still believe that “The future of direct cardiac reprogramming: any GMT cocktail variety?” is an attractive title and the one that best reflects the content of this review. “Cocktail” has a metaphoric meaning, and the final question mark is the key feature. The reader can evaluate after reading the review if there is a clear answer to this question or continue doubting whether the future of direct cardiac reprogramming relies on any of the described GMT-based cocktail varieties. Anyhow, if the reviewer thinks that there could be a better title for this review, we will take it gladly into consideration.

2. Authors should provide an illustration or a section that discusses the advantages or limitations of direct reprogramming vs cell-based therapeutics or endogenous CM proliferation response. We have introduced at the beginning of section 2 of the revised manuscript a paragraph (lines 347-368 in track changes mode) that describes the main advantages of direct cardiac reprogramming as a strategy for cardiac regeneration compared to the use of other cells and over cell-based therapy in the case of in vivo cardiac reprograming. On the other hand, the current major limitations of direct cardiac reprogramming are mentioned throughout the discussion (section 3).

3. One of the biggest limitations with direct reprogramming is the low efficiency of conversion into CMs. Authors should discuss this in a separate section together with discussion of the current literature that targets enhancement of CM generation. A table representation may be helpful as well. In order to clearly reflect that one the major limitations of direct cardiac reprogramming is the low efficiency and that most  studies  have aimed to increase conversion efficiency, we have included in the revised manuscript a paragraph after the first description of the GMT cocktail (lines 386-388 in track changes mode). Moreover, following suggestions from the reviewer, we have included an extra column in the Table 1 to describe the reprogramming efficiency of all the studies described in the review.

4. First 2 sections, "Cardiogenesis" and "Regenerative medicine" can be shortened. We believe that sections 1.2 and 1.3 are necessary for the reader to understand the current major limitations of regenerative medicine in the cardiac field and why novel strategies (such as direct cardiac reprogramming) are needed. Moreover, following the second reviewer´s suggestions, we have rewritten section 1.1 “Cardiogenesis”, to describe cardiogenesis in relation to direct cardiac reprogramming. To do this, we have rearranged some paragraphs in sections 1.1 and 1.2 and have changed the title of section 1.3. More importantly, we have provided information about genes that play a fundamental role in cardiogenesis, and that have been used in most of the direct cardiac reprogramming strategies towards iCMs and iCPCs. Thus, the reader can relate to and understand better the rationale for having selected and used these genes in the reprogramming cocktails described throughout section 2.

5. It may helpful if the authors could provide a table showing the impact of every direct reprogramming approach on cellular functions such as survival, proliferation, and cardiac commitment. Following the helpful advice of this reviewer, we have included an extra column in the Table 1 to describe the impact of every direct reprogramming approach on the functionality of iCMs and iCPCs obtained.

Reviewer 2 Report

This article discusses rapidly emerging a new research field, direct cardiac reprogramming, which has brought a particular attention in regenerative medicine as an alternative approach to cell replacement therapy.  A series of review papers on this topic have been published for last several years.  Given that new progresses have been made in this field at a very rapid pace, this topic will be still one of great interests of readers of International Journal of Molecular Medicine.   The suggestions are below.

1.1 Cardiogenesis:  It would be helpful for the readers to understand direct cardiac reprogramming if the authors focus on cardiac transcriptional networks which not only drives cardiomyogenesis, but also was a mechanistic basis of direct cardiac reprogramming.  The author should describe cardiogenesis in relation to direct cardiac reprogramming.

Page 2 first paragraph: epicardium should be included several layers of the heart.

1.3 A brief overview on the current methods used to improve cell survival:

The title and the content do not seem to be matched.  It would be better to describe the current progresses and challenges on cell therapy for heart diseases.

  1. Direct reprogramming for heart regeneration:

Given that the authors described cell therapy in the previous section, it is necessary to discuss the main rationale for developing direct cardiac reprogramming for heart regeneration between section 1.3 and 2 or in the beginning of section 2 (e.g. direct cardiac reprogramming targets endogenous cardiac cells, thus it may bypass the obstacles associated with cell therapy as described).

2.1. Direct cardiac reprogramming in vitro: The authors listed the different reprogramming protocols.  It could be restructured to describe how the researchers have modified the original cocktail to improve the efficiency.  It can be classified: 1) additional transcription factors, 2) miRNA, 3) chemicals (i.e. notch inhibition, Tgf-beta inhibition, prostaglandin signaling inhibition, and etc), 4) epigenetic modification (i.e Bmi1), and 5) modifying stoichiometry

2.1.1.  Direct reprogramming into mouse iCMs: It is worthy of discussing the requirement of Hand2 for efficient reprogramming as demonstrated by multiple studies: 1) Int. J. Mol. Sci. 2017, 18, 1805; doi:10.3390/ijms18081805, and 2) Scientific Reports | (2019) 9:6362 | https://doi.org/10.1038/s41598-019-42945-w, and 3) Scientific Reports | (2019) 9:6362 | https://doi.org/10.1038/s41598-019-42945-w

Table 1. there are a few recent studies missing: 1) NATURE COMMUNICATIONS | (2019) 10:674 | https://doi.org/10.1038/s41467-019-08626-y, 2) Int. J. Mol. Sci. 2017, 18, 1805; doi:10.3390/ijms18081805, 3) Scientific Reports | (2019) 9:6362 | https://doi.org/10.1038/s41598-019-42945-w, 4) Scientific Reports | (2019) 9:14970 | https://doi.org/10.1038/s41598-019-51536-8,

2.2. Direct cardiac reprogramming in vivo

Third paragraph: the study using MGT polycistronic vector is missing (Cardiovasc Res

2015 Nov 1;108(2):217-9.doi: 10.1093/cvr/cvv223. Epub 2015 Sep 23)

Page 24, 1st paragraph: Song et al. used Tcf21MerCreMer:Rosa26tdTomato mice in addition to FSP1-Cre: Rosa26 LacZ mice for lineage tracing.

Author Response

This article discusses rapidly emerging a new research field, direct cardiac reprogramming, which has brought a particular attention in regenerative medicine as an alternative approach to cell replacement therapy.  A series of review papers on this topic have been published for last several years.  Given that new progresses have been made in this field at a very rapid pace, this topic will be still one of great interests of readers of International Journal of Molecular Medicine.  The suggestions are below.

1.1 Cardiogenesis:  It would be helpful for the readers to understand direct cardiac reprogramming if the authors focus on cardiac transcriptional networks which not only drives cardiomyogenesis, but also was a mechanistic basis of direct cardiac reprogramming.  The author should describe cardiogenesis in relation to direct cardiac reprogramming. We sincerely thank the reviewer for this helpful advice.  We have rewritten this section, to describe cardiogenesis in relation to direct cardiac reprogramming as suggested. To do this, we have rearranged some paragraphs in sections 1.1 and 1.2 and more importantly, we have provided information about genes that play a fundamental role in cardiogenesis and comment briefly on some of the transcriptional networks these factors establish and their function. Finally, we have introduced a final sentence at the end of this section “As described further below, the aforementioned genes have been considered and used in direct cardiac reprogramming strategies towards CM- and CPC-like states.”  In this way, the reader can relate to and understand better the rationale for having selected and used these genes in the reprogramming cocktails described throughout section 2.

Page 2 first paragraph: epicardium should be included several layers of the heart. The epicardium as the outermost layer of the pericardium has been included in the text (line 49 and 50 in track changes mode). 

1.3 A brief overview on the current methods used to improve cell survival:

The title and the content do not seem to be matched.  It would be better to describe the current progresses and challenges on cell therapy for heart diseases. We agree with the reviewer and the title of 1.3 section has been changed accordingly.

 Direct reprogramming for heart regeneration:

Given that the authors described cell therapy in the previous section, it is necessary to discuss the main rationale for developing direct cardiac reprogramming for heart regeneration between section 1.3 and 2 or in the beginning of section 2 (e.g. direct cardiac reprogramming targets endogenous cardiac cells, thus it may bypass the obstacles associated with cell therapy as described). We totally agree with the reviewer. Following this suggestion, we have introduced at the beginning of section 2 of the revised manuscript a paragraph (lines 347-368 in track changes mode) that describes the main advantages of direct cardiac reprogramming as a strategy for cardiac regeneration compared to the use of other cells and over cell-based therapy in the case of the in vivo cardiac reprograming.

 2.1. Direct cardiac reprogramming in vitro: The authors listed the different reprogramming protocols.  It could be restructured to describe how the researchers have modified the original cocktail to improve the efficiency.  It can be classified: 1) additional transcription factors, 2) miRNA, 3) chemicals (i.e. notch inhibition, Tgf-beta inhibition, prostaglandin signaling inhibition, and etc), 4) epigenetic modification (i.e Bmi1), and 5) modifying stoichiometry. We are again grateful for the reviewer´s advice. We also consider that this point can improve the clarity of the text and text flow. Accordingly, we have introduced a brief title to each paragraph of section 2.1 to describe the type of strategy used to increase reprogramming efficiency as indicated by the reviewer.

 2.1.1.  Direct reprogramming into mouse iCMs: It is worthy of discussing the requirement of Hand2 for efficient reprogramming as demonstrated by multiple studies: 1) Int. J. Mol. Sci. 2017, 18, 1805; doi:10.3390/ijms18081805, and 2) Scientific Reports (2019) 9:6362 | https://doi.org/10.1038/s41598-019-42945-w, and 3) Scientific Reports (2019) 9:6362 | https://doi.org/10.1038/s41598-019-42945-w . We agree with the reviewer and we have described these studies in section 2.1.1 “Direct reprogramming into mouse iCMs” (Modifications to the GMT cocktail-Inclusion of additional transcription factors: the relevance of Hand2 transcription factor). Moreover, a full paragraph has been included in the section 3 “Future directions and challenges” (lines 811 to 820 in track changes mode).

 Table 1. there are a few recent studies missing: 1) NATURE COMMUNICATIONS | (2019) 10:674 | https://doi.org/10.1038/s41467-019-08626-y, 2) Int. J. Mol. Sci. 2017, 18, 1805; doi:10.3390/ijms18081805, 3) Scientific Reports (2019) 9:6362 | https://doi.org/10.1038/s41598-019-42945-w, 4) Scientific Reports (2019) 9:14970 | https://doi.org/10.1038/s41598-019-51536-8. We very much appreciate the reviewer´s observation. These studies have been described in the main text (section 2.1.1 Direct reprogramming into mouse iCMs) and included in the Table 1 in the revised version.

 2.2. Direct cardiac reprogramming in vivo

Third paragraph: the study using MGT polycistronic vector is missing (Cardiovasc Res 2015 Nov 1;108(2):217-9.doi: 10.1093/cvr/cvv223. Epub 2015 Sep 23). This study was already included in this section in the original version of the manuscript (lines 675 to 678 in track changes mode, Ma et al., reference 144).

Page 24, 1st paragraph: Song et al. used Tcf21MerCreMer:Rosa26tdTomato mice in addition to FSP1-Cre: Rosa26 LacZ mice for lineage tracing. Tcf21MerCreMer:Rosa26tdTomato mice have been included in the text (line 707 in track changes mode) in the revised version.

Round 2

Reviewer 1 Report

All concerns have been addressed

Author Response

We would like to thank this reviewer for the time and effort invested in reviewing this article to improve its quality.

Reviewer 2 Report

The revised version of manuscript is responsive to the reviewer's comments.  However, the description of epicardium is still incorrect.  The heart wall is composed of three layers including endocardium, myocardium, and epicardium which is a outer layer.  Pericardium is basically the membrane wrapping around the heart and is not directly connected to the heart wall.  It is separated by pericardial fluid from the heart.

Author Response

We agree with the reviewer, we apologize for this mistake. We have corrected the epicardium information in the text accordingly (lines 47-50, track changes mode). We would like to thank this reviewer for the time and effort invested in reviewing this article to improve its quality.